# Unveiling the Role of Exosomes in the Pathophysiology of Sepsis: Insights into Organ Dysfunction and Potential Biomarkers

**DOI:** 10.3390/ijms25094898

**Published:** 2024-04-30

**Authors:** Gizaw Mamo Gebeyehu, Shima Rashidiani, Benjámin Farkas, András Szabadi, Barbara Brandt, Marianna Pap, Tibor A. Rauch

**Affiliations:** 1Institute of Biochemistry and Medical Chemistry, Medical School, University of Pécs, 7624 Pécs, Hungary; mamo_gizaw@yahoo.com (G.M.G.); sh.rashidiani@gmail.com (S.R.); farkas.beni97@gmail.com (B.F.); 2Department of Dentistry, Oral and Maxillofacial Surgery, Medical School, University of Pécs, 7623 Pécs, Hungary; szabadi.andras@pte.hu; 3Hungary Department of Medical Biology and Central Electron Microscope Laboratory, Medical School, University of Pécs, 7624 Pécs, Hungary; bb.barbara91@gmail.com (B.B.); marianna.pap@aok.pte.hu (M.P.)

**Keywords:** sepsis, vital organ dysfunction, circulating exosomes, pathophysiology, molecular biomarkers, septic shock

## Abstract

Extracellular vesicles (EVs) are tools for intercellular communication, mediating molecular transport processes. Emerging studies have revealed that EVs are significantly involved in immune processes, including sepsis. Sepsis, a dysregulated immune response to infection, triggers systemic inflammation and multi-organ dysfunction, posing a life-threatening condition. Although extensive research has been conducted on animals, the complex inflammatory mechanisms that cause sepsis-induced organ failure in humans are still not fully understood. Recent studies have focused on secreted exosomes, which are small extracellular vesicles from various body cells, and have shed light on their involvement in the pathophysiology of sepsis. During sepsis, exosomes undergo changes in content, concentration, and function, which significantly affect the metabolism of endothelia, cardiovascular functions, and coagulation. Investigating the role of exosome content in the pathogenesis of sepsis shows promise for understanding the molecular basis of human sepsis. This review explores the contributions of activated immune cells and diverse body cells’ secreted exosomes to vital organ dysfunction in sepsis, providing insights into potential molecular biomarkers for predicting organ failure in septic shock.

## 1. Introduction

Sepsis is a serious medical condition that can result in life-threatening organ dysfunction if left untreated [1]. It is crucial to recognize the symptoms and understand its progression, underlying pathology, and available treatment options. Early symptoms of sepsis can initially present with non-specific symptoms such as fever, chills, rapid breathing, increased heart rate, and confusion. As sepsis progresses, symptoms may include a significant drop in blood pressure (septic shock), difficulty breathing, decreased urine output, altered mental status, and organ dysfunction. If not promptly treated, sepsis can progress to severe sepsis or septic shock, where multiple organs fail due to inadequate blood flow and oxygen delivery. Sepsis is triggered by an infection, which may be bacterial, fungal, or viral. The infection causes the release of inflammatory mediators (such as cytokines) that activate the immune system excessively [2]. Excessive inflammation can lead to microvascular dysfunction, tissue damage, and impaired organ function. In addition, the dysregulation of coagulation pathways can occur, leading to disseminated intravascular coagulation (DIC), which further complicates the condition. Treatment options are very limited; therefore, it is crucial to promptly administer broad-spectrum antibiotics in order to target the underlying infection. Intravenous fluids are given to restore adequate blood flow and blood pressure [3]. In cases of septic shock, medications such as norepinephrine may be needed to raise blood pressure. Patients frequently require oxygen therapy, mechanical ventilation, and renal replacement therapy (dialysis) to support failing organs. In some cases, corticosteroids may be employed to modulate the immune response. Sepsis is a leading cause of mortality and multiple organ failure in intensive care units [4]. It is a dysregulated systemic inflammatory response triggered by pathogen-associated molecular patterns (PAMPs) and damage-associated molecular patterns (DAMPs) upon infection, which is mediated through pattern recognition receptors (PRRs) [5]. Despite intensive research, deciphering the intricate inflammatory pathogenesis underlying sepsis-induced organ dysfunction remains a challenge. Recent efforts have focused on understanding the role of secreted exosomes in complex pathological processes (Figure 1), aiming to identify biomarkers associated with their multifaceted pathophysiology [6,7,8,9,10,11]. Consequently, sepsis is a medical emergency that requires early recognition and aggressive treatment, thus underscoring the unmet need for new therapies.

## 2. Biogenesis of Exosomes and Cargo Packaging

EVs are classified into three main categories based on size, membrane markers, biogenesis, and release pathways. These categories are microvesicles, exosomes, and apoptotic bodies [12]. The secretion of EVs containing miRNA is not a random or collateral event (as occurs with apoptotic bodies) but, rather, an event that is desired by the cells to facilitate intercellular communication [13]. Recent research suggests that each subgroup includes multiple subpopulations, which likely have distinct biological roles and different effects on recipient cells [14,15]. Exosomes are smaller in diameter (30 to 150 nm) compared to microvesicles. They form through the inward budding of intracellular endosomes within the cytoplasm, leading to the creation of multivesicular bodies (MVBs) [6,16]. These MVBs ultimately release exosomes into the extracellular space following fusion with the cell membrane (Figure 1) [15].

Exosomes, depending on their cargo, regulate cellular homeostasis by affecting signaling pathways and enzyme reactions, thereby modulating the function and cellular phenotype of the recipient cells [17,18]. However, it is important to note that the biochemical contents of exosomes can vary significantly depending on the cell type, pathological conditions, and environment [19]. The metabolic and functional state of the recipient cells can also play a role in determining the biological effects of exosomes [20]. Exosomes transfer their functional cargo to recipient cells through different mechanisms, such as endocytosis, membrane fusion, or phagocytosis. Studies suggest that exosomes found in bodily fluids may contain molecules related to diseases, making them potential biomarkers for various human diseases such as sepsis, cancers, neurodegenerative diseases, and autoimmune disorders [6,7,17]. Additionally, the composition and concentration of exosomes in body fluids are directly correlated with the pathophysiological state of the originating cells [4,10].

Exosomes contain several characteristic protein markers, including CD63, CD9, and CD81, as well as specific membrane components such as cholesterol, sphingomyelin, and phosphatidylinositol [21]. Exosomes carry a complex array of membrane-associated proteins, oligomeric proteins, functional mRNA, miRNA, DNA fragments, and lipids, indicating their potential functional diversity [12]. The proper sorting and packaging of cargo into exosomes represents a fundamental process in the biogenesis and function of exosomes. The ESCRT (Endosomal Sorting Complex Required for Transport) machinery is a collection of protein complexes (ESCRT-0, ESCRT-I, ESCRT-II, and ESCRT-III) that are involved in the formation of ILVs within MVBs. Cargo sorting and packaging into ILVs primarily rely on the ESCRT machinery, which recognizes and sequesters specific cargo proteins into budding vesicles. ESCRT complexes recognize ubiquitinated proteins on the endosomal membrane and facilitate their sorting into ILVs. The ESCRT machinery interacts with specific protein and lipid components to promote ILV formation and cargo encapsulation. On the other hand, ESCRT-independent pathways involve alternative mechanisms for cargo sorting and vesicle formation. Lipid-based microdomains, tetraspanin-enriched domains, and other protein complexes can contribute to cargo sorting and vesicle budding. Cargo selection in ESCRT-independent pathways is likely to involve lipid raft domains, specific protein–protein interactions, or RNA-mediated sorting. ESCRT-independent vesicle budding can occur through membrane curvature induced by specific lipid–protein interactions or through the action of other protein complexes. It is indisputable that alternative membrane remodeling proteins (e.g., SPG7) contribute to vesicle formation and release. ESCRT-dependent pathways are characterized by post-translational modifications, including ubiquitin-mediated cargo recognition and selective sorting. ESCRT-independent pathways, on the other hand, may rely on lipid microdomains, protein–protein interactions, or RNA-based mechanisms for cargo selection. In conclusion, ESCRT-dependent and ESCRT-independent mechanisms represent alternative pathways for cargo sorting and vesicle formation during exosome biogenesis. Understanding these pathways is essential for elucidating how exosomes communicate with cells and contribute to disease processes (Figure 2). Selective packaging of microRNAs (miRNAs) and messenger RNAs (mRNAs) into exosomes involves intricate mechanisms that govern cargo sorting and loading [22]. Certain motifs or structural features within miRNAs can influence their selective packaging into exosomes. Exosome-associated RNA-binding proteins and lipid components may recognize specific sequences or secondary structures of miRNAs. RNA-binding proteins, such as hnRNPA2B1, Ago2, and others, interact with miRNAs and facilitate their loading into exosomes [23]. Components of ESCRT and related proteins play roles in sorting miRNA into exosomes. The ESCRT machinery can interact with RNA-binding proteins and specific membrane domains to facilitate the encapsulation of miRNAs into intraluminal vesicles (ILVs) within multivesicular bodies (MVBs). Specific RNA-binding proteins and complexes, such as ELAVL1 (HuR), hnRNPs, and Ago2, interact with mRNAs and regulate their sorting into exosomes [24]. These proteins can recognize mRNA sequences, localization signals, or secondary structures, determining their inclusion in the exosome cargo. The ESCRT machinery and related components contribute to sorting mRNAs into ILVs during exosome biogenesis [25]. Understanding these processes is crucial for elucidating the roles of exosomal RNAs in intercellular communication, disease pathogenesis, and potential therapeutic applications.

## 3. Significantly Altered Exosomal Cargo in Sepsis

During sepsis, immune and non-immune cells become activated and release exosomes extensively [10,26]. Plasma exosomes in sepsis patients have been extensively studied, revealing significant differences in their biochemical composition, functions, and circulation levels compared to those in a healthy state [7,27]. During sepsis, altered exosomes exhibit elevated levels of pro-inflammatory, anti-inflammatory, and pro-coagulant properties (Table 1).

## 4. Alteration in Cell Metabolism

Cellular metabolism involves both anabolic and catabolic processes that maintain physiological equilibrium and produce essential chemical energy (ATP) for cellular functions [54]. In septic cells, there is a heightened demand for glucose to fulfill their biosynthetic and bioenergetic needs [55]. This alteration occurs as cells switch from oxidative phosphorylation to glycolysis, which is mediated by hypoxia-inducible factor-1α (HIF-1α). As a result, this change disrupts the normal balance between catabolism and anabolism, contributing to the pathophysiology of sepsis and the manifestation of clinical symptoms [56]. Sepsis induces the production of exosomes that contain HIF-1α, which has a profound effect on cytokine production, cellular metabolism, and adaptation [56]. Additionally, studies have shown significantly increased levels of HIF-1 mRNA in exosomes isolated from the blood of patients with septic shock [56]. Research suggests that HIF derived from exosomes has potential as a promising biomarker for sepsis [6,56,57].

Exosomes deliver metabolites (e.g., glucose, amino acids, lipids, and nucleotides) in addition to proteins, lipids, and nucleic acids to recipient cells, influencing various aspects of cellular metabolism [58]. Exosomes can also transfer glucose transporters, enzymes, and regulators involved in glucose metabolism [59]. Delivered lipids and lipid-modifying enzymes can influence lipid metabolism and storage in recipient cells. Transfer of metabolites via exosomes can supplement cellular energy production, biosynthesis pathways, and other metabolic processes in recipient cells. Exosomal miRNAs can target specific genes involved in metabolic regulation, thereby altering cellular metabolism in recipient cells [60]. Further research into exosome-mediated metabolic regulation will deepen our understanding of cellular communication and metabolic homeostasis.

## 5. Endothelial and Cardiovascular Dysfunctions

Endothelial cells play a crucial role in regulating vascular function, including the production of nitric oxide (NO), a key signaling molecule involved in vascular homeostasis [61]. Exosomes released by endothelial cells can contribute to intercellular communication and may impact NO production and vascular function through various mechanisms. Endothelial cells produce NO through the action of endothelial nitric oxide synthase (eNOS), which converts L-arginine into NO and citrulline [62]. NO is a potent vasodilator that regulates blood vessel tone, inhibits platelet aggregation, and promotes endothelial integrity. Exosome-derived miRNAs regulate eNOS expression and activity in recipient cells [63]. Exosomal lipid components can also modulate eNOS function or NO release in target cells [64]. Strategies to enhance NO’s bioavailability or restore endothelial cell function via exosome-based therapies are being explored for cardiovascular diseases [65]. When vascular endothelial cells encounter PAMPs, such as lipopolysaccharides (LPSs), they activate various inflammatory mediators through PRRs, inducing a state of both pro- and anti-inflammatory imbalance in sepsis [4]. This dysregulation significantly contributes to septic shock and organ dysfunction, and it ultimately serves as a predictor of mortality in sepsis [30,66,67]. Detecting early-stage endothelial and cardiovascular dysfunctions could be crucial for implementing effective control measures during the initial phase of sepsis.

Research has revealed the multiple roles of exosomes derived from various cells (such as monocytes, platelets, erythrocytes, neutrophils, and endothelial cells) that are activated during sepsis and influence both the physiology and pathophysiology of endothelial cells [8,28]. Some of the effects of these exosomes include promoting excessive nitric oxide production, lesion formation, intravascular calcifications, plaque progression, inflammation, and coagulation. Conversely, other components within exosomes contribute to vascular protection and endothelial regeneration [28,68,69]. Therefore, exosomes present a promising avenue for therapeutics, potentially serving as a novel tool for detecting and managing endothelial and cardiovascular dysfunctions [68].

## 6. Coagulation Disorders

Disorders in blood coagulation in sepsis are influenced by exosomes, which act as crucial messengers in inflammatory signaling through cell-to-cell communication [10]. These exosomes contribute to the three primary characteristics of coagulation disorders in sepsis: activation of coagulation, disruption of anticoagulant systems, and imbalances in fibrinolytic systems [10]. In sepsis, exosomes derived from platelets containing pro-coagulant elements such as phosphatidylserine and tissue factors have been found to significantly contribute to activating coagulation by orchestrating the assembly of blood-clotting enzyme complexes and initiating the coagulation cascade [10,68]. Recent research suggests that exosomes from different cell types, such as leukocytes, endothelial cells, and red blood cells, play important roles in mediating coagulation disorders during sepsis by modulating both pro- and anti-inflammatory reactions [28,70]. Numerous studies have elucidated the relationship between various exosomal components and sepsis-induced coagulopathy [10,20,28,68,70]. These studies have highlighted specific exosomal components strongly associated with sepsis-induced coagulation disorders, including tissue factors, phosphatidylserine, neutrophil extracellular traps (DNA and histone), damage-associated molecular patterns (such as chromosomal DNA, nucleosome, mitochondrial DNA, high-mobility group box 1 protein, and heat shock proteins), complements, messenger RNA, and microRNA.

## 7. Exosome Circulation and Its Implications in Sepsis-Induced Vital Organ Dysfunction

Several studies have described the pathophysiological mechanisms that cause sepsis, which are mainly characterized by dysregulated inflammatory responses and oxidative stress resulting from the activation of PRRs on recipient cells. The activation takes place immediately after the binding of the PAMPs and DAMPs [4,71,72,73]. PAMPs, such as LPSs found in the outer membrane of Gram-negative bacteria, are essential structures for microbial pathogenicity [74]. Conversely, DAMPs are molecules released during inflammatory stress or from dying cells during sepsis [72]. PRRs are ubiquitous cell surface receptors expressed in various cell types, including immune effector cells, endothelial cells, epithelial cells, and myocytes [75]. These receptors chiefly recognize intricate immune molecules. Toll-like receptors (TLRs), nucleotide-binding oligomerization domain (NOD)-like receptors, retinoic acid-inducible gene (RIG)-like receptors, mannose-binding lectins, and scavenger receptors are receptors that initiate various signaling pathways involved in inflammation, adaptive immunity, and cellular metabolism during sepsis [75]. The uncontrolled inflammatory responses and oxidative stress result in vascular endothelial dysfunction, cellular metabolic alterations, and coagulation irregularities. These factors collectively lead to multi-organ dysfunction [71,76,77]. Studies on sepsis have highlighted changes in the contents and functions of exosomes. Exosomes transport high levels of cytokines and DAMPs, such as tissue factors, nucleosomes, mitochondrial DNA, high-mobility group box 1, heat shock proteins, histones, adenosine triphosphate, extracellular RNA, and phosphatidylserine, which play crucial roles in inducing systemic inflammation and thrombogenesis [6,10,78,79]. Research on cell lines exposed to LPS, animal models injected with LPS, and human septic patients has shown that sepsis-induced exosomes contain increased levels of inflammatory cytokines and chemokines such as IL-1β, IL-2, IL-6, IL-12, IL-15, IL-17, TNF-α, and IFN-γ, which promote the synthesis of secondary mediators, migration of inflammatory cells, and collateral tissue damage [8,28,80]. Additionally, these studies indicate the presence of anti-inflammatory cytokines, such as IL-4 and IL-10, in exosomes [81]. These cytokines modulate immunosuppressive pathways in later stages, which may lead to imbalances between pro-inflammatory and anti-inflammatory states. Several studies have highlighted numerous miRNAs within septic exosomes from patients, LPS-treated macrophages, or septic mice. These miRNAs regulate dysregulation of the inflammatory response by targeting proteins in inflammatory signaling pathways such as toll-like receptors, P38-mitogen activated protein kinase (MAPK), and necrosis factor-kappa β (NF-κB) [6,28]. Comparisons between exosomes from septic patients and healthy individuals revealed differential expression of various miRNAs associated with disease severity and mortality [28,30,34,35,37,38,39,40,41,42,43]. Moreover, in sepsis, exosomes have distinct compositions of miRNAs, mRNAs, proteins, and lipids compared to their healthy counterparts. These differences can have significant pathological effects on the lungs, kidneys, liver, cardiovascular system, and central nervous system, which can lead to consequential injuries (Figure 2).

### 7.1. Central Nervous System (CNS)

Septic shock and septic encephalopathy are among the primary causes of impaired consciousness in patients with sepsis [82]. Sepsis-related encephalopathy is characterized by diffuse cerebral dysfunction resulting from the systemic inflammatory response to infection. It can occur independently of other causes, such as liver or renal dysfunction, even without direct infection in the CNS [6,82]. Exosome-related information from the CNS is primarily available from animal models. Studies in mice treated with LPS indicate that exosomes are released and present in significant quantities in the cerebrospinal fluid, suggesting their role in molecular transport across the blood–brain barrier during sepsis [83,84]. Moreover, certain exosome components, such as miR-146a and miR-155, stimulate microglia and astrocytes, leading to the secretion of pro-inflammatory cytokines and other factors (Figure 2). This contributes to inflammation, oxidative stress, and nitric oxide production in the brain [83,84,85]. Exosomes, whether circulating or released locally, can induce brain inflammation, which directly contributes to encephalopathy and may cause systemic immune dysfunction [6]. In the context of sepsis, exosomes released from immune cells, such as microglia and astrocytes in the brain, may contribute to the pathogenesis and progression of neurological complications associated with sepsis. These exosomes can transport pro-inflammatory cytokines, damage-associated molecular patterns (DAMPs), and microRNAs, which can induce neuroinflammation, disrupt the blood–brain barrier, and impair neuronal function [86,87,88]. Conversely, some research indicates that exosomes may also have neuroprotective effects during sepsis. They can carry anti-inflammatory molecules, neurotrophic factors, and other molecules that promote neuronal survival and repair [89]. Additionally, exosomes derived from stem cells or other therapeutic sources show potential for mitigating sepsis-induced brain injury by modulating immune responses, promoting tissue repair, and enhancing neuronal resilience [90]. Overall, the role of exosomes in the brain during sepsis is complex and multifaceted. Further research is needed to elucidate the specific mechanisms by which exosomes influence neurological outcomes in septic patients, as well as to explore their potential as therapeutic targets or agents for neuroprotection in this context.

### 7.2. Cardiovascular System

In the heart and vascular system, exosomes participate in intercellular communication and homeostasis under normal physiological conditions, in addition to playing a role in mediating inflammation and tissue injury during sepsis [91]. Exosomes derived from cardiac cells, such as cardiomyocytes and cardiac fibroblasts, play essential roles in cardiac remodeling, angiogenesis, and the regulation of cardiac function [92]. These exosomes carry signaling molecules, such as growth factors (e.g., VEGF, FGF), microRNAs, and proteins involved in extracellular matrix remodeling, which modulate cellular responses and contribute to cardiac adaptation to stress (Figure 2) [93]. Furthermore, endothelial-derived exosomes are critical for maintaining vascular integrity, regulating vascular tone, and mediating intercellular crosstalk between endothelial cells and other vascular cell types [94]. These exosomes can transfer bioactive molecules, including nitric oxide, endothelial nitric oxide synthase (eNOS), and angiogenic factors, which modulate vascular function and angiogenesis [95]. Heart failure and dysfunction are significant causes of death in septic patients. Sepsis-induced cardiomyopathy is a major contributor to septic shock, along with hypovolemia [96]. This condition results from various factors, including pro-inflammatory mediators, mitochondrial dysfunction, oxidative stress, altered calcium regulation, abnormal autonomic nervous activity, and endothelial dysfunction [97,98]. Recent research suggests that exosomes may contribute to cardiac damage during sepsis [6]. Elevated levels of exosomes derived from platelets, leukocytes, and endothelial cells have been observed in sepsis patients, as well as in chronic vascular diseases such as atherosclerosis, and they exacerbate inflammation and microvascular permeability [53,80]. Studies on exosome cargoes from septic patients have highlighted increased levels of activities such as nicotinamide adenine dinucleotide phosphate (NADPH) oxidase, leading to the production of reactive oxygen species (ROS) and reactive nitrogen species (RNS), causing oxidative stress [97,99]. Exosomes also carry nitric oxide (NO) and peroxynitrite, which induce myocardial dysfunction in isolated heart and muscle preparations [100,101]. In sepsis-induced cardiomyopathy models, in vivo experiments suggest that inhibiting exosome release can improve cardiac function, indicating a potential clinical impact on this condition [102]. During sepsis, the balance between pro-inflammatory and protective effects of exosomes in the cardiovascular system may be disrupted, leading to endothelial dysfunction, microvascular thrombosis, and myocardial depression. Understanding the dynamic interplay between different populations of exosomes and their cargoes in the context of sepsis-induced cardiovascular dysfunction is crucial for developing targeted diagnostic and therapeutic strategies. Additionally, exosomes hold promise as potential biomarkers for the early detection and monitoring of sepsis-associated cardiovascular complications [103]. Analysis of exosomal cargo, such as specific microRNAs or proteins associated with cardiac injury and inflammation, may provide valuable insights into the pathophysiology of sepsis-induced cardiac dysfunction and help guide clinical decision-making. In summary, exosomes play diverse and intricate roles in both physiological and pathological processes in the heart and vascular system, including their involvement in sepsis-induced cardiovascular dysfunction. Further research into the mechanisms underlying exosome-mediated communication and their therapeutic potential is warranted to improve the management of sepsis and its associated cardiovascular complications.

### 7.3. Lungs

Exosomes have emerged as important mediators of pulmonary inflammation and injury in the context of sepsis (Figure 2). During sepsis, a dysregulated immune response to infection can lead to acute lung injury (ALI) or its more severe form, acute respiratory distress syndrome (ARDS). During sepsis, the lungs are highly susceptible to systemic inflammatory dysregulation, which can result in ALI or ARDS [104]. This vulnerability often occurs alongside multiple organ failure, particularly acute kidney injury, which further increases sepsis-associated mortality [104]. In sepsis, inflammatory molecules such as DAMPs and PAMPs can negatively impact the alveolar–capillary barrier, leading to pulmonary edema, fluid influx, and subsequent lung injury [105]. Recent studies in septic patients and animals have highlighted the significant roles of exosomes and other extracellular vesicles in both promoting and reducing the inflammatory processes of sepsis in the lungs [106]. For example, when wild-type mice were treated with exosomes from septic human patients, the expression of endothelial nitric oxide synthase (eNOS), extracellular superoxide dismutase (SOD), cyclooxygenase-2 (COX-2), and NF-κB in the heart and lungs increased compared to treatment with exosomes from healthy controls [80]. Additionally, eCIRP has been identified in exosomes isolated from septic mice or patients and has been linked to the induction of acute ALI in sepsis [107]. Exosomes produced by pulmonary structural cells, immunoregulatory cells, and stem cells were found to be increased in the bronchoalveolar lavage fluid (BALF) of infectious ALI mice [108]. These exosomes have been shown to significantly contribute to lung inflammation in various ALI models [108]. Exosomes and microvesicles loaded with miR-155 were secreted in the BALF and serum of LPS-challenged macrophages. This secretion induced expression of TNF-α and IL-6 in the lungs, caused lung endothelial cell apoptosis, and disrupted the alveolar–capillary barrier [109]. The effects of serum exosomes from septic patients on LPS-induced pulmonary dysfunction in animal models have drawn considerable attention for proposing novel therapeutic approaches to managing sepsis-induced multiple organ dysfunctions [79]. Exosomes derived from mesenchymal stem cells (MSCs) or other therapeutic sources possess anti-inflammatory, immunomodulatory, and tissue repair properties. These exosomes can attenuate pulmonary inflammation, enhance alveolar fluid clearance, and promote tissue regeneration, thereby mitigating lung injury and improving survival in experimental models of sepsis-induced ALI/ARDS [110]. Overall, exosomes play a complex and multifaceted role in the pathogenesis of pulmonary complications during sepsis. Further research is needed to elucidate the specific mechanisms by which exosomes contribute to sepsis-induced lung injury, and to explore their potential as diagnostic biomarkers or therapeutic targets in sepsis-associated pulmonary diseases. Understanding the dynamic interplay between different populations of exosomes and their cargoes in the lung microenvironment may provide new insights into the pathophysiology of sepsis and facilitate the development of novel strategies for the prevention and treatment of sepsis-induced lung injury.

### 7.4. Liver

Exosomes secreted by various liver cell types, including hepatocytes, immune cells, and endothelial cells, play crucial roles in intercellular communication and the regulation of immune responses in the liver (Figure 2). During sepsis, a systemic inflammatory response to infection, the liver plays a central role in host defense and the clearance of pathogens and their byproducts. Exosomes have emerged as important mediators of hepatic inflammation and injury in the context of sepsis. However, they have also been implicated in inducing septic liver dysfunction due to elevated bilirubin concentrations and the occurrence of coagulation disorders [111,112]. This dysfunction directly contributes to the genesis and exacerbation of dysfunction in other vital organs during sepsis [111,113]. Elucidating the pathophysiology and clinical manifestations of sepsis-associated liver dysfunction could standardize diagnostic panels for early and precise diagnosis. Studies have shown that exosomes and other EVs released from macrophages, neutrophils, hepatocytes, and liver sinusoidal endothelial cells are involved in sepsis-associated liver dysfunction [114,115,116]. For example, in mice, exosomes from macrophages challenged with LPS were taken up by hepatocytes, which subsequently stimulated the production and secretion of mature interleukin-1 beta (IL-1β) and IL-18 through NLRP3 inflammasome and caspase-1 activation [117,118]. This process resulted in the infiltration of macrophages and neutrophils, as well as elevated serum levels of AST, ALT, and LDH [117,118]. Research on human and animal models has indicated that exosomes released by various hepatic cells, including hepatocytes and Kupffer cells, contain DAMPs such as HMGB1 and HSP90. These activate TLRs and subsequently upregulate the expression of pro-inflammatory genes in Kupffer cells [117,119,120]. Studies on exosomal miR-155 have identified its role as a regulator of inflammation, targeting multiple components of the pro-inflammatory cytokine production cascade in the liver [27]. Its upregulation in inflammatory conditions or in the liver under LPS and/or TLR9 ligand stimulation suggests its involvement in liver dysfunction [27]. Exosomal miR-103-3p from LPS-activated macrophages targets Krüppel-like factor 4 (KLF4), increasing the expression of α-SMA, TGF-β, and Col1a1 in hepatic stellate cells, contributing to chronic liver dysfunction or liver fibrosis post-sepsis [121]. It is challenging, if not impossible, to differentiate exosomes derived from sepsis from those originating from previous liver damage or sepsis-induced liver damage. The impact of altered cargo, including miRNAs and proteins in exosomes and other extracellular vesicles during sepsis, on acute and chronic liver dysfunction remains an area requiring further exploration [122]. Furthermore, exosomes released from injured or stressed hepatocytes may exacerbate liver injury by inducing apoptosis, oxidative stress, and hepatic stellate cell activation [123]. These exosomes can also carry microRNAs and other molecules that regulate gene expression and contribute to the pathogenesis of liver dysfunction during sepsis. Conversely, some studies suggest that certain populations of exosomes may have protective effects on the liver during sepsis. Exosomes derived from mesenchymal stem cells (MSCs) or other therapeutic sources possess anti-inflammatory, immunomodulatory, and tissue repair properties. These exosomes can attenuate hepatic inflammation, reduce hepatocyte apoptosis, and promote tissue regeneration, thereby mitigating liver injury and improving survival in experimental models of sepsis-induced liver dysfunction [123].

### 7.5. Kidneys

Exosomes have emerged as important mediators of renal inflammation and injury in the context of sepsis (Figure 2). Exosomes derived from immune cells, such as macrophages, neutrophils, and dendritic cells, contribute to the propagation of inflammation in the kidneys during sepsis [124]. These exosomes carry pro-inflammatory cytokines, chemokines, and damage-associated molecular patterns (DAMPs) that activate renal epithelial cells, resident immune cells, and endothelial cells, leading to the recruitment of inflammatory cells and the amplification of the immune response. During sepsis, the development of acute kidney injury (AKI) is a common and serious complication that can lead to increased morbidity and mortality [125]. Exosomes containing cytokines are produced by different types of cells, such as immune cells, endothelial cells, and tubular epithelial cells, when their PRRs are activated by PAMPs or DAMPs during sepsis in the context of AKI [6,125]. Exosomes containing cytokines have been identified as direct mediators that trigger kidney inflammation in sepsis. Studies on sepsis patients and animal models have highlighted the pivotal roles of exosomes’ prothrombotic, pro-inflammatory, and immunomodulatory properties in the development of AKI. For example, exosomes released by macrophages challenged with LPS contain histones, which act as DAMPs and induce inflammatory responses that damage kidney tubular cells [6,125,126]. The exosomes induced by sepsis have RNA contents, including miRNA19b-3p, which can activate specific pathways and promote inflammation, contributing significantly to the development of AKI [6]. Exosomes found in urine and circulation are potential targets for treating sepsis-induced AKI due to their involvement in organ damage [127]. Certain populations of exosomes may have protective effects on the kidneys during sepsis. Exosomes derived from mesenchymal stem cells (MSCs) or other therapeutic sources possess anti-inflammatory, immunomodulatory, and tissue repair properties. These exosomes can attenuate renal inflammation, reduce tubular cell apoptosis, and promote tissue regeneration, thereby mitigating kidney injury and improving survival in experimental models of sepsis-induced AKI. Exosomes have garnered interest as potential diagnostic biomarkers for sepsis-induced renal dysfunction [128].

## 8. Biomedical Application of Sepsis-Induced Exosomes

Mesenchymal stem cells (MSCs) are a type of adult stem cell found in various tissues, including bone marrow, adipose tissue, and umbilical cord blood [129]. Mesenchymal stem cells (MSCs) have gained considerable attention in the field of regenerative medicine due to their distinctive characteristics. They have the capacity to differentiate into various cell types, regulate immune responses, and facilitate tissue repair. These properties make them a promising candidate for therapeutic applications. In the context of sepsis, MSCs have shown promise as a potential therapy for mitigating the dysregulated immune response and tissue damage associated with this life-threatening condition. MSCs exert their therapeutic effects through various mechanisms, including in a paracrine manner (e.g., immunomodulation, anti-inflammatory effects, and tissue repair) [50]. These effects stem back to the secretion of anti-inflammatory cytokines (e.g., IL10), transforming growth factor-beta (TGF-β), and indoleamine 2,3-dioxygenase (IDO), which can dampen the inflammatory cascade and promote tissue healing. Among these paracrine mediators, exosomes have generated significant interest due to reports of their therapeutic effects. Stem cell exosomes can potentially avoid the safety risks involved in the delivery of cell therapies. The biomedical application of exosomes induced by sepsis presents promising prospects in both the diagnostic and therapeutic realms (Figure 3) [130]. Exosomes, which are small extracellular vesicles, have significant potential as biomarkers for sepsis due to their disease-dependent composition and concentration in biofluids [130]. Their cargo, which contains sepsis-related information, makes them valuable indicators for diagnosis and prognosis in septic conditions [131,132]. Recent research has emphasized the therapeutic advantages of exosomes from mesenchymal stem cells (MSCs) in reducing sepsis-induced organ dysfunction [133]. These exosomes transport RNAs and proteins to target cells, demonstrating potential therapeutic effectiveness in preventing sepsis-related complications [134]. MSC exosomes have a safer profile compared to their parent cells, can be stored without losing their function, and possess other advantageous characteristics [135].

Exosomes have diverse application prospects in sepsis, such as early diagnosis, dynamic disease monitoring, targeted therapy, and potential utilization as a vaccine platform for sepsis prevention [135,136]. However, to fully characterize their biological functions and establish their viability as a treatment option in clinical settings [137], it is crucial to further develop and optimize methods for exosome isolation. Although there are promising applications, there are still several unanswered questions about the precise functions of exosomes during sepsis, under both physiological and pathological conditions. It is essential to address these knowledge gaps to fully harness the potential of exosomes in managing sepsis and improving patient outcomes.

## 9. Future Direction of MSC Exosome Therapeutic Approach

MSC exosome (MSC-Exo) therapy, which involves the use of exosomes derived from mesenchymal stem cells, has the potential to impact various aspects of future medicine, including the following:

1. Precision Medicine: With advancements in our understanding of the mechanisms of action and refining isolation techniques, MSC-Exo therapy could become more personalized. Tailoring exosome payloads to target specific diseases or individual patient characteristics may enhance treatment efficacy and minimize adverse effects.

2. Diverse Therapeutic Applications: MSC-Exo therapy might find applications across a wide range of medical conditions beyond those already explored. These could include neurological disorders, cardiovascular diseases, autoimmune conditions, and even cancer. Clinical trials and preclinical research may uncover novel therapeutic avenues for MSC-Exo in diverse medical fields.

3. Regulatory Approval and Standardization: As research progresses and more clinical evidence accumulates, regulatory agencies may establish guidelines for the development and approval of MSC-Exo-based therapies. Standardization of isolation methods, characterization techniques, and quality control measures will be crucial for ensuring safety, efficacy, and reproducibility.

4. Combination Therapies: MSC-Exo therapy could be integrated with other treatment modalities such as gene editing, drug delivery systems, or tissue engineering strategies to enhance therapeutic outcomes. Synergistic effects between MSC-Exo and complementary therapies could lead to more robust regenerative responses or targeted interventions.

5. Non-Invasive Delivery Routes: Innovations in delivery methods could enable non-invasive administration of MSC-Exo, such as inhalation, topical application, or targeted delivery using nanoparticles or biomaterial scaffolds. These approaches may improve patient compliance, reduce procedural risks, and enable targeted delivery to specific tissues or organs.

6. Long-Term Monitoring and Follow-Up: Longitudinal studies will be essential to assess the long-term safety, durability, and potential side effects of MSC-Exo therapy. Monitoring patient outcomes over extended periods will provide insights into the persistence of therapeutic effects, potential immunogenicity, and any risks associated with repeated administration.

7. Cost-Effectiveness and Accessibility: As manufacturing processes mature and economies of scale are realized, MSC-Exo therapy may become more cost-effective and accessible to a broader patient population. Efforts to optimize production workflows, reduce manufacturing costs, and streamline regulatory pathways could contribute to broader adoption and affordability.

8. Bioengineering and Biomimicry: Advances in bioengineering techniques may enable the design of synthetic exosome mimetics or engineered MSCs optimized for exosome production. These engineered systems could offer improved scalability, customization, and control over therapeutic payloads, opening up new possibilities for precision medicine and regenerative therapies.

Overall, the future of MSC-Exo therapy holds tremendous potential for revolutionizing regenerative medicine and addressing unmet medical needs across various clinical domains. However, realizing this potential will require continued interdisciplinary collaboration, robust clinical evidence, regulatory support, and technological innovation [138].

## 10. Exosomes for Pathogen-Directed Therapy

As per its definition, sepsis is a host overreaction to a microbial infection that causes damage to the body itself. In the preceding sections, our focus has been on host-directed therapies that aim to minimize the damage caused by sepsis, including the mitigation of the process and tissue regeneration. The development of new drugs and delivery methods is crucial in addressing the issue of antibiotic resistance. It is worth considering that by using MSC-based exosomes, it may be possible to target the infectious agents [139,140]. Exosomes can be modified or engineered to express targeting molecules, such as peptides or antibodies that can recognize and bind to microbial pathogens [141,142]. This targeting can enable the delivery of therapeutic cargo directly to infected cells or pathogens, which can minimize off-target effects and maximize efficacy [143]. On the other hand, it is noteworthy that exosomes can be loaded with antimicrobial agents such as antimicrobial peptides, small molecules, or nucleic acid-based therapeutics (e.g., miRNAs) that specifically target essential pathways or components of the pathogen [144]. By utilizing these antimicrobial compounds, we can disrupt microbial growth, replication, or virulence, resulting in clearance or inhibition of the pathogen. The synergistic effects between exosomes and antimicrobial drugs can potentially potentiate their antimicrobial activity, reduce the development of drug resistance, and improve overall therapeutic efficacy. Overall, exosome therapy directed towards pathogens has emerged as a promising strategy to combat microbial infections, offering targeted and versatile approaches to enhance antimicrobial efficacy while minimizing adverse effects on host tissues. However, it is important to note that further research and development efforts are needed to fully realize the potential of this innovative therapeutic approach.

## 11. Challenges in the Field of Extracellular Vesicle Research

Although extracellular vesicles, including exosomes and microvesicles, offer great promise, they do present a number of challenges for researchers in the field. These challenges can affect various aspects of exosome research, including the following:

1. One of the primary challenges in exosome research is the development of robust and reproducible methods for isolating and purifying exosomes from biological fluids or cell cultures. Current isolation techniques, such as ultracentrifugation, size-exclusion chromatography, and polymer-based precipitation methods, often have limitations in terms of yield, purity, and preservation of exosome integrity.

2. Exosome heterogeneity: Exosomes represent a heterogeneous population of vesicles that vary in size, cargo, and cellular origin. This heterogeneity presents challenges in standardizing exosome isolation and characterization protocols.

3. Exosome characterization: Comprehensive analysis of exosomes is required to ascertain their size distribution, morphology, surface markers, and cargo contents (proteins, lipids, nucleic acids). However, many current techniques for EV characterization, such as electron microscopy, nanoparticle tracking analysis, and flow cytometry, have limitations in sensitivity, specificity, and reproducibility.

4. Cargo profiling and functional studies are essential for understanding the cargo carried by exosomes, including microRNAs (miRNAs), messenger RNAs (mRNAs), and proteins. These studies are crucial for elucidating the roles of exosomes in intercellular communication and disease pathogenesis.

5. Standardization and quality control: The absence of standardized protocols for exosome isolation, characterization, and functional assays impedes the reproducibility and comparability of research findings across different studies. The establishment of guidelines and quality control measures for exosome research is crucial for advancing the field and translating EV-based therapies to clinical applications.

6. The biological complexity of exosomes and their in vivo applications present significant challenges. Factors such as biodistribution, clearance mechanisms, and immune responses to exosomes must be thoroughly investigated to ensure the safety and efficacy of EV-based therapies.

The challenges associated with extracellular vesicle research can be addressed through interdisciplinary collaboration among scientists, engineers, and clinicians. Additionally, concerted efforts should be made to establish standardized protocols, quality control measures, and regulatory frameworks for exosome-based therapies.

## 12. Conclusions

During the progression of sepsis, exosomes released from activated cells play a crucial role in facilitating cellular communication and impacting key pathophysiological mechanisms that underlie vital organ dysfunctions. These mechanisms include coagulation and thrombosis, angiogenesis, oxidative stress, immune modulation, and inflammation. However, although exosomes have been acknowledged to be involved, their internal and external molecular contents and their relation to the multifaceted pathophysiologies of vital organ dysfunctions during sepsis remain largely unexplored. Comprehensive future investigations, in both preclinical and clinical settings, are necessary to address this knowledge gap and elucidate the potential regulatory influence of exosomal contents on the primary pathophysiological events characterizing sepsis. Combining host- and pathogen-directed exosome therapy with conventional antimicrobial agents or other therapeutic modalities may enhance treatment outcomes. Such research endeavors hold promise for unveiling novel therapeutic strategies aimed at intervening in the progression of sepsis, potentially reducing mortality rates and hospitalizations associated with this critical condition.

## Figures and Tables

**Figure 1 ijms-25-04898-f001:**
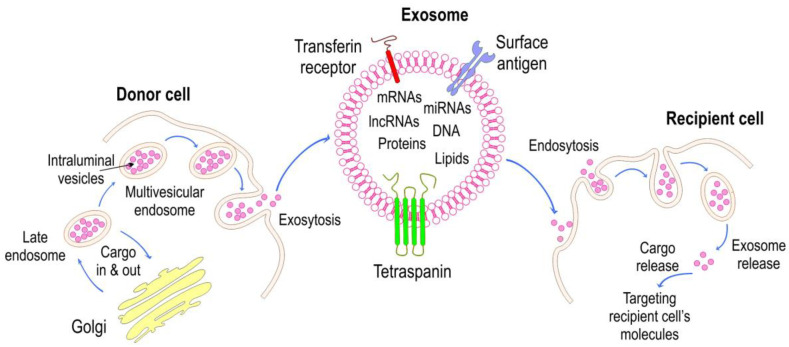
Exosome biogenesis and their diverse molecular contents: Exosome-releasing multivesicular bodies (MVBs) are formed in donor cells through a multistage process involving endocytosis through the plasma membrane, formation of early and late endosomes, enrichment of exosomes with specific membrane proteins, and packaging with molecular cargo such as mRNAs, miRNAs, proteins, and lipids during MVB maturation. Ultimately, MVBs fused to the donor cell membrane and exosomes are released into the extracellular space. Exosomes can enter recipient cells through specific protein–protein interactions, such as ligand–receptor binding, or by direct fusion with the plasma membrane. Once inside the host cell, exosomes release their macromolecular cargo into the cytoplasm, affecting signaling pathways and translation. They can also generate regulatory proteins, such as transcription factors, which switch specific target genes on or off.

**Figure 2 ijms-25-04898-f002:**
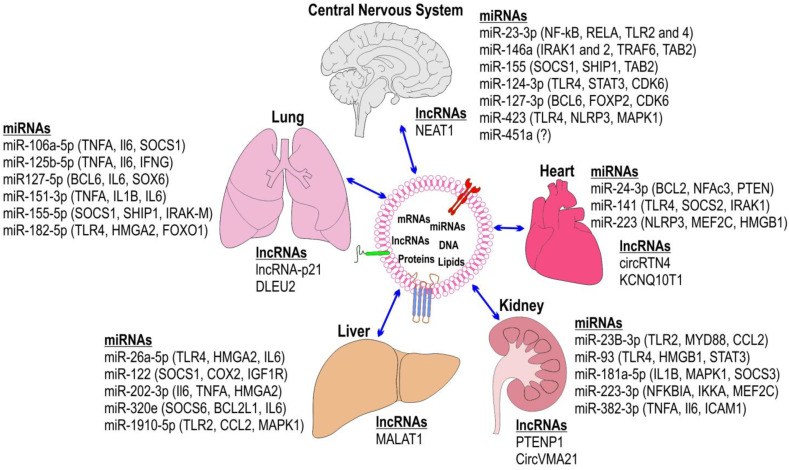
Upregulated exosomal RNA cargoes in sepsis: MiRNAs and long non-coding RNAs (lncRNA), including circular RNAs, are listed next to the organs in which they were discovered. MiRNA target genes are listed next to the corresponding miRNA. The organ-specific sections (7.1–7.5) provide more details and protein factors of exosome cargoes.

**Figure 3 ijms-25-04898-f003:**
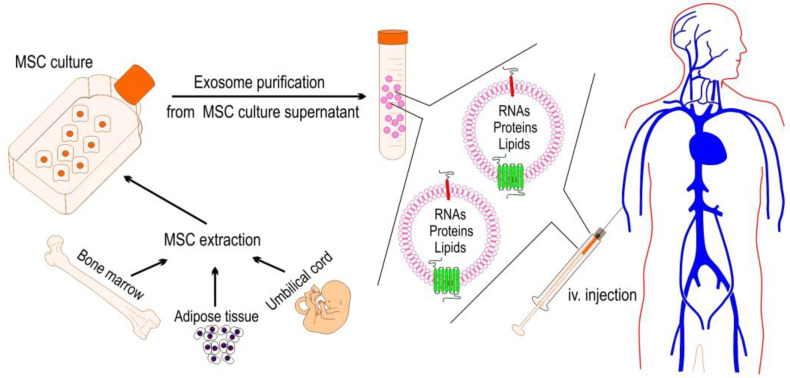
Therapeutic approach based on exosomes derived from mesenchymal stem cells. The process involves isolating and culturing MSCs, followed by administering purified exosomes via intravenous (i.v.) injection. These exosomes have appropriate signals on their membrane, which enable them to locate target tissues and cells in the human body.

**Table 1 ijms-25-04898-t001:** Exosomal cargo, including proteins, mRNAs, miRNAs, and lncRNAs, in sepsis. Target genes and the resulting pathophysiological changes are included.

**Source/** **Model**	**Exosomal Cargo**	**Expression in Sepsis**	**Target Gene**	**Pathophysiological Change**	**Ref.**
	Proteins				
Human	Variety of cytokines/ chemokines	Up	ND	Modulate inflammation by regulating target proteins in inflammatory signaling pathways	[28,29]
Human	HSPs	Up	ND	Work as DAMPs to induce inflammation	[28]
Human	HMGB1	Up	ND	Mediates the release of inflammatory factors via acting on immune cells, pyroptosis pathways, and phosphorylating nuclear factor-kB	[28]
Human	SPTLC3	Up	ND	Involved in sphingolipid metabolism, with a negative correlation with the progression of sepsis	[30]
Mice, Human	ATF3	Up	*Noxa and Bnip3*	An early diagnostic biomarker for sepsis-induced acute kidney injury	[31]
Human	Histones	Up	CXCL9 and CXCL10	Specifically target monocytes in human blood, which evokes the mobilization of the chemotactic chemokines from these cells	[32]
Human	NADPH oxidase, NO synthase	Up	ND	Induce endothelial cell apoptosis and regulation of sepsis-induced metabolic alterations	[6]
	mRNAs				
Human	MPO, PRDX3, SOD2, FOXM1, SELS, and GLRX2	Up	ND	Regulation of sepsis-induced oxidative stress	[33]
Contd…					
Human	DNMT1, DNMT3A, DNMT3B	Up	ND	Regulate gene expression by modifying DNA methylation and altering transcription	[30]
	miRNAs				
Human	miR-1-3p, miR-21-3p, miR-221-3p, mirR-129-5p, miR-222-3p, miR-221-5p, miR-155-5p, miR-1247-3p, miR-148a-5p, and miR-222-5p	Up	SERP1, SORBS2, BAK1, P53, PTEN	Induce endothelial cell dysfunction and regulates sepsis-related cardiac dysfunction through modulating target genes	[28,30,34,35]
Human	et-7b-5p, let-7c-5p, miR-122-5p, miR-1227-3p	Up	ND	Modulate inflammatory signaling pathways and the cell cycle by regulating target proteins in sepsis	[6,23,28]
Mice	miR-16, miR-17, miR-20a, miR-20b, miR-26a, miR-26, miR-106a, miR-106b, miR-195, miR-451	Up	SIRPa, CDK6, cyclin E1, IFN-b	Regulate macrophage infiltration, phagocytosis, pro-inflammatory cytokine secretion, and G1/S-phase progression by targeting specific genes	[36]
Mice	miR-19a, miR-21, miR-27a, miR-126, miR-146b, miR-200	Up	TNF-α and IL-17A	Induce immunosuppression in sepsis by suppressing target genes’ expression	[29]
Human	miR-125b-5p, miR-21-5p, miR-30a-5p, miR-100-5p, miR-122-5p, miR-193a-5p,	Up	ND	Induce myocardial dysfunction and apoptosis in endothelial cells, and may thus contribute to the vascular abnormalities commonly observed in patients with sepsis	[37]
Mice	miR-23b	Up	NF-κB, IL-17	Regulates the NF-κB-mediated activation of vascular endothelial cells	[38]
Mice	miR-126-3p, miR-122-5p, miR-146a-5p, miR-145-5p, miR-26a-5p, miR-150-5p, miR-222-3p, miR-181a-5p	Up	TLR7-MyD88	Mediate the cytokine production	[39,40]
Mice	miR-34a-5p, miR-122-5p, miR-145-5p, miR-146a-5p, miR-210-3	Up	IL-6, TNF-a, IL-1b, and MIP-2	Induce complement activation, cytokine production, and leukocyte migration in sepsis	[39]
Mice	miR-499-5p	Up	EIF4E	Able to regulate sepsis-induced cardiomyopathy by targeting EIF4E	[41]
Human	miR-122	Up	ND	*Levels correlated with short-term mortality in sepsis patients*	[42]
Contd…					
Mice	miR-193b	Down	NF-κB p65/HDAC3	Systemic exosomal miR-193b-3p delivery attenuates the inflammatory response by acetylation of the NF-κB p65 via suppressed expression and activity of HDAC3	[43]
Mice	miR-181b	Down	NF-kB	Regulates NF-kB-mediated endothelial cell activation and vascular inflammation in response to pro-inflammatory stimuli	[44]
Human	miR-21	Up		Involved in the regulation of late sepsis	[29]
Human	miR-155	Up	ND	Positive regulator of inflammation by virtue of its potent upregulation in multiple immune cell lineages by TLR, inflammatory cytokines, and specific antigens	[45]
Human	miR-15a	Up	ND	Upregulated miR-15a downregulates the LPS-induced inflammatory pathway	[46]
Mice	miR-574-5p	Up	ND	A good predictor for sepsis prognosis	[47]
Human	miR-133a	Up	ND	Correlation between the levels of miR-133a and sepsis severity	[48]
	lncRNAs				
Human	TUG1	Down	miR-142-3p/Sirtuin 1 axis and NF-kB	Decreasing its expression may contribute to the development of sepsis-associated acute kidney injury via modulating target genes	[29]
Human	TapSAKI	Up	miR-22/ PTEN/TLR4/NF-kB	Promotes inflammation injury in HK-2 cells	[49]
Human Mice	MALAT1	Up	SAA3,miR-12b	TNF-a expression in LPS-induced septic cardiomyocytes via activation of target gene	[50]
Mice	Hotairm1	Up	S100A	Support of myeloid-derived suppressor cells’ expansion during sepsis	[51]
Human	NEAT1	Up	miR-22-3p	The upregulation of NEAT1 was related to the severity of acute kidney (AKI) in sepsis	[52]

These properties (i.e., cargo) directly contribute to pathophysiological processes, including changes in cell metabolism, endothelial dysfunction, coagulation disorders, and cardiovascular dysfunction [6,28,53].

## Data Availability

Not applicable.

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
