# Peer review of "Unveiling the Role of Exosomes in the Pathophysiology of Sepsis: Insights into Organ Dysfunction and Potential Biomarkers"

_ijms, 2024, doi:10.3390/ijms25094898_

Round 1

Reviewer 1 Report

Comments and Suggestions for Authors

Unveiling the Role of Exosomes in Sepsis Pathophysiology: Insights into Organ Dysfunction and Potential Biomarkers

     Comments-

1.     The author has not explained Sepsis properly in the Introduction part, nor has included any data on patients suffering from it in recent years. In the introduction, the author should also mention what treatments have been administered for this disease so far. Overall, there is a need to rewrite the introduction section more effectively.

2.     The part about exosomes biogenesis has been written very lightly, making it difficult to understand the process of exosomes biogenesis. Here, the author should mention which pathways (such as ESCRT-dependent and ESCRT-independent) are involved in exosomes biogenesis and which protein complexes are involved in them. Doing so will improve the quality of the biogenesis paragraph. I would suggest to cite the Karn et al., 2021 (https://www.mdpi.com/2227-9059/9/10/1373) paper.  

3.     In paragraph “Alteration in cell metabolism” including additional information about other bioactive molecules associated with exosomes and how they alter metabolism can make this paragraph even more informative.

4.     In the paragraph about endothelial and cardiovascular dysfunctions, the author has explained that exosomes derived from various cells activated during sepsis influence both the physiology and pathophysiology of endothelial cells. Some effects of these exosomes include promoting excessive nitric oxide production, lesion formation, intravascular calcifications, plaque progression, inflammation, and coagulation. The author needs to mention here the proteins, miRNA, or other bioactive molecules associated with these exosomes that play a significant role in carrying out these functions. This would facilitate understanding of the mechanisms through which these functions occur.

5.     In Figure 2, diverse types of miRNA are depicted across various organ systems. Nevertheless, the manuscript lacks elucidation regarding the regulatory dynamics of these miRNAs within the organs, whether they are up-regulated or down-regulated, and their specific targets such as proteins or pathways. Emphasizing this aspect would enhance the clarity and comprehensibility of this paragraph.

6.     I would suggest authors to include the separate section for the challenges in the field of extracellular vesicles research.  

The oveall, the paper is very interesting. 

Comments on the Quality of English Language

The minor English editing is required.  

Author Response

Authors’ answer to the Reviewer:

We do appreciate the reviewer's time spent reviewing our manuscript and the constructive comments and suggestions. We have incorporated the suggested additions and discussions to make the manuscript clearer. More specifically:

Reviewer’s comment #1: The author has not explained sepsis properly in the Introduction part, nor has included any data on patients suffering from it in recent years. In the introduction, the author should also mention what treatments have been administered for this disease so far. Overall, there is a need to rewrite the introduction section more effectively.

Authors’ answer: According to the reviewer’s suggestions, we rewrite the Introduction section.

Reviewer’s comment #2: The part about exosomes biogenesis has been written very lightly, making it difficult to understand the process of exosomes biogenesis. Here, the author should mention which pathways (such as ESCRT-dependent and ESCRT-independent) are involved in exosomes biogenesis and which protein complexes are involved in them. Doing so will improve the quality of the biogenesis paragraph. I would suggest to cite the Karn et al., 2021 (https://www.mdpi.com/2227-9059/9/10/1373) paper.

Authors’ answer: We considered this remark with great care and determined that it would be beneficial to expand the section by providing a more detailed description of exosome biogenesis. We have added the suggested paper to the reference list.

Reviewer’s comment #3: In paragraph “Alteration in cell metabolism” including additional information about other bioactive molecules associated with exosomes and how they alter metabolism can make this paragraph even more informative.

Authors’ answer:  We have added additional information to the manuscript to make this section more informative.

Reviewer’s comment #4: In the paragraph about endothelial and cardiovascular dysfunctions, the author has explained that exosomes derived from various cells activated during sepsis influence both the physiology and pathophysiology of endothelial cells. Some effects of these exosomes include promoting excessive nitric oxide production, lesion formation, intravascular calcifications, plaque progression, inflammation, and coagulation. The author needs to mention here the proteins, miRNA, or other bioactive molecules associated with these exosomes that play a significant role in carrying out these functions. This would facilitate understanding of the mechanisms through which these functions occur.

Authors’ answer: We are grateful for the reviewer’s input and are discussing his/her ideas in more detail regarding the action and regulation of nitric oxide.

Reviewer’s comment #5:  In Figure 2, diverse types of miRNA are depicted across various organ systems. Nevertheless, the manuscript lacks elucidation regarding the regulatory dynamics of these miRNAs within the organs, whether they are up-regulated or down-regulated, and their specific targets such as proteins or pathways. Emphasizing this aspect would enhance the clarity and comprehensibility of this paragraph.

Authors’ answer: The listed RNAs, including miRNAs, circRNAs, and long non-coding RNAs, are upregulated in the context of sepsis. The targeted gene names added after the corresponding miRNAs in Figure 2.

Reviewer’s comment #6: I would suggest authors to include the separate section for the challenges in the field of extracellular vesicles research.  

Authors’ answer: Thank you for your suggestions; we have added a new section in the revised manuscript to address this question.

Reviewer 2 Report

Comments and Suggestions for Authors

The authors consider sepsis management to be critical from a clinical perspective.

- A distinction between EVs should be considered, as it seems increasingly clear that the secretion of those containing miRNA is not a random or collateral event (as happens for apoptotic bodies) but an event desired by the cells to obtain inter-cellular communication (10.1158/0008-5472.CAN-14-2568).

- One of the limitations of miRNAs is that they are related to different proteins and genetic pathways, so this should be analyzed and distinguished. ECVs can be related, for example, to diabetes (10.3390/ph14121257) or liver diseases (10.3390/cimb45010006), which could be present but not directly related to sepsis

The therapeutic approach can be interesting but still has important limitations. It would be necessary to identify one or a group of specific proteins but, above all, to identify a target. What could it be in sepsis?

So, although the approach is promising, it should be better characterized; I would suggest distinguishing between the two possible options, apoptotic bodies, and cytokines or miRNAs

Comments on the Quality of English Language

It needs revision.

Author Response

Authors’ answer to the Reviewer

We do appreciate the reviewer's time spent reviewing our manuscript and the constructive comments and suggestions. We have incorporated the suggested additions and discussions to make the manuscript clearer. More specifically:

Reviewer’s comment #1: A distinction between EVs should be considered, as it seems increasingly clear that the secretion of those containing miRNA is not a random or collateral event (as happens for apoptotic bodies) but an event desired by the cells to obtain inter-cellular communication (10.1158/0008-5472.CAN-14-2568).

Authors’ answer: We are grateful for this observation and have incorporated it into the manuscript. The suggested paper has been cited in accordance with the relevant citation style.

Reviewer’s comment #2: One of the limitations of miRNAs is that they are related to different proteins and genetic pathways, so this should be analyzed and distinguished. ECVs can be related, for example, to diabetes (10.3390/ph14121257) or liver diseases (10.3390/cimb45010006), which could be present but not directly related to sepsis.

Authors’ answer: Thank you for your remarks. We do know that it is challenging, if not impossible, to differentiate exosomes derived from sepsis from those originating from previous liver damage or sepsis-induced liver damage. Anyway, we have taken your suggestion into account and incorporated it into the manuscript. We have also cited the suggested papers.

Reviewer’s comment #3: The therapeutic approach can be interesting but still has important limitations. It would be necessary to identify one or a group of specific proteins but, above all, to identify a target. What could it be in sepsis?

Authors’ answer: We have devoted a new section in the revised manuscript to address these questions.

Reviewer’s comment #4: So, although the approach is promising, it should be better characterized; I would suggest distinguishing between the two possible options, apoptotic bodies, and cytokines or miRNAs

Authors’ answer: A substantial increase in the discussion of the advantages and disadvantages in the revised manuscript may satisfy the reviewer.

Round 2

Reviewer 2 Report

Comments and Suggestions for Authors

I appreciated the hard work made by the authors, the manuscript improved a lot!

Comments on the Quality of English Language

Fine just some revision needed